# A New Proposed Symbiotic Plant–Herbivore Relationship between *Burkea africana* Trees, *Cirina forda* Caterpillars and Their Associated Fungi *Pleurostomophora richardsiae* and *Aspergillus nomius*

**DOI:** 10.3390/microorganisms11071864

**Published:** 2023-07-24

**Authors:** Lufuno Ethel Nemadodzi, Gerhard Prinsloo

**Affiliations:** 1Department of Agriculture and Animal Health, University of South Africa, Private Bag X6, Johannesburg 1710, South Africa; 2ABBERU, Science Campus, University of South Africa, Johannesburg 1710, South Africa

**Keywords:** endophytes, *Pleurostomophora richardsiae*, *Aspergillus nomius*, lepidoptera, mutual benefit, plant-herbivore interaction

## Abstract

*Burkea africana* is a tree found in savannah and woodland in southern Africa, as well as northwards into tropical African regions as far as Nigeria and Ethiopia. It is used as fuel wood, medicinally to treat various conditions, such as toothache, headache, migraine, pain, inflammation, and sexually transmitted diseases, such as gonorrhoea, but also an ornamental tree. The current study investigated the possible symbiotic relationship between *B. africana* trees and the *C. forda* caterpillars and the mutual role played in ensuring the survival of *B*. *africana* trees/seedlings in harsh natural conditions and low-nutrient soils. Deoxyribonucleic acid isolation and sequencing results revealed that the fungal species *Pleurostomophora richardsiae* was highly predominant in the leaves of *B*. *africana* trees and present in the caterpillars. The second most prominent fungal species in the caterpillars was *Aspergillus nomius*. The latter is known to be related to a *Penicillium* sp. which was found to be highly prevalent in the soil where *B. africana* trees grow and is suggested to play a role in enhancing the effective growth of *B*. *africana* trees in their natural habitat. To support this, a phylogenetic analysis was conducted, and a tree was constructed, which shows a high percentage similarity between *Aspergillus* and *Penicillium* sp. The findings of the study revealed that *B. africana* trees not only serve as a source of feed for the *C. forda* caterpillar but benefit from *C. forda* caterpillars which, after dropping onto the soil, is proposed to inoculate the soil surrounding the trees with the fungus *A. nomius* which suggests a symbiotic and/or synergistic relationship between *B. africana* trees and *C. forda* caterpillars.

## 1. Introduction

*Burkea africana* (Fabaceae: Caesalpinioideae: Caesalpinieae) trees occur in various types of woodland over a wide range of altitudes and habitats but are most characteristic in hot, low-lying areas [1]. The trees are deciduous, and the leaves fall from May to September, and new leaves flush from August to December [2]. Flowers appear from August to November, whereas fruit ripens from February to October and can remain on the tree for a long time [1,3]. In South Africa, *B. africana* is prevalent in Mpumalanga, Limpopo, and parts of Gauteng Provinces. There have been numerous attempts at growing *B*. *africana* trees outside their natural habitat through the excavation of seedlings or the raising of seedlings after the germination of seeds. However, it has been reported that seedlings only survive for 6–8 months after removal from their natural habitat [4]. *Burkea africana* trees grow in clusters in their natural habitat, also indicating selectivity in conditions to support growth and further development of seedlings [4]. 

*Burkea africana* is well known as an important host of the caterpillars of the Caster semi-looper moth *Cirina forda* (Lepidoptera: Bombycoidea; Saturniidae)*,* with infestation occurring annually from November to January (Figure 1). Primary hosts also include *Vitellaria paradoxa* (shea Butter tree), *Euclea divinorum*, *Acacia mearrnsii*, *Manilkara sulcata*, and *Crossopteryx febrifuga* [5,6] for the caterpillars *C. forda*, with *B*. *africana* being the most common and preferred in South Africa.

*Cirina forda* caterpillars are widely used as food in Africa, especially in Nigeria, Zimbabwe, Zambia, South Africa, Central Africa, and the Democratic Republic of Congo [7]. In South Africa, the caterpillars are commonly known as “mashonzha” and are considered a delicacy among the VhaVenda, BaPedi, and VaTsonga, which mainly reside in the Limpopo Province [4]. Recently, *C. forda* has gained popularity among the AmaNdebele people, which is found mainly in Mpumalanga Province. The larvae are handpicked, squeezed to remove the entrails, boiled in salty water for longer preservation, dried, and sold at the local markets. It is then prepared as a relish served with porridge made of maise meal. Current studies do not report any significant health problems associated with the consumption of edible insects, including *C. forda,* and are therefore considered safe for consumption. Several studies have been conducted on the nutritional health benefits offered by edible insects indicating that such insects contain sufficient amounts of good-quality protein and other important nutrients [8,9,10]. *Cirina forda* caterpillars are no exception, known to be high in crude protein and vitamins. 

The fungal species, particularly *P. richardsiae* and *A. nomius* species, have been reported in the leaves and nuts of different trees. *Pleurostomophora richardsiae* was initially known as a human pathogen [11,12,13,14] as the cause of subcutaneous phaeohyphomycotic cysts after traumatic implantation [15]. Currently, *P. richardsiae*, *P. repens, Pl. ootheca* [16] and *P. ochracea* [17] are the four species that are recognised. *Aspergillus nomius* was first described in 1987 [18] and is reported to be the producer of both B and G-type aflatoxins [19]. *Aspergillus nomius* has been identified in Pistachio nuts [20], wheat [18], maise, and peanuts [19] and in agricultural soils in the US [18,21], Iran [22] and Thailand [23]. 

No published Information is available regarding the potential role of *C. forda* in the growth and development of *B. africana* trees. Although several studies have been conducted on the medicinal properties of *B. africana*, to date, no information is available on why the caterpillars are selectively attracted to *B. africana* trees to feed on the leaves. A previous study clearly showed the differences in the soil around the trees (*Burkea* soils) in comparison to other soils (non-*Burkea* soils), using soil metabolomic analysis. The nutrient content, plant growth regulating compounds, as well as microorganism differentiation was described by Nemadodzi et al. [4]. The aim of the current study was to elucidate the possible symbiotic relationship that exists between the caterpillars and the trees and their potential role in the growth and establishment of *B. africana* trees in their natural environment to explain and support the soil metabolomics analysis as reported by Nemadodzi et al. [4]. This is the first study to report on the presence of two fungal species found in or on the leaves of *B. africana* trees and *C. forda* caterpillars (Figure 1) and the suggested co-dependent symbiotic relationship between the caterpillars and the trees. 

## 2. Materials and Methods

### 2.1. Collection of Leaf and Caterpillar Sample

Newly developed leaves were harvested in October 2017 from randomly selected *B. africana* trees at Telperion Game Reserve, which covers approximately 1000 ha and is situated in Mpumalanga province, South Africa. *Cirina forda* caterpillars were randomly collected from these trees by handpicking them from the leaves and the ground surrounding *B. africana* trees in November 2017. The study was conducted at three different sites within the reserve, namely site 1 (S 25°42′40.00″; E 029°00′21.6″), site 2 (S 25°41′26.6″; E 029°01′46.7″) and site 3 (S 25°39′49.4″; E 029°01′59.7″). Telperion is situated in the summer rainfall region of South Africa, with annual rainfall ranging from 570–730 mm [24]. According to Brown et al. [25], the average temperature indicates February to be the hottest month of the year, with 26.4 °C as the average daily maximum, whilst 15.1 °C is the average daily minimum. The collection period of September–November falls within the spring and summer seasons, with temperatures ranging from a maximum of 24 °C and a minimum of 12 °C.

### 2.2. Genomic DNA PCR and Sequencing

Fresh leaves of *B. africana* were harvested at Telperion Game Reserve and placed in brown bags, and stored at −80 °C until use. A total of 30 live caterpillars were handpicked, and the intestines were squeezed out and put in enclosed bottles which were stored at −80 °C to limit microbial contamination. Both the leaves and caterpillars were sent to Inqaba Biotechnical Industries, a commercial service provider, for next-generating sequencing (NGS) for the identification of differences in a mixed microbial species [26] and/or population in a sample through purifying and sequencing following the protocol below: 

ITS Metagenomics: (V3) regions were amplified in a 25 uL reaction using Q5^®^ Hot start High-Fidelity 2× Master Mix (New England Biolabs, Ipswich, MA, USA). Amplicon library PCR was performed on all replicate extractions separately. The DNA primers used were Truseq-tailed ITS 1F and ITS 4. Thermocycler settings for PCR amplification were as follows: (1) initial denaturation at 95 °C for 2 min (2) 30 cycles of 95 °C for 20 s (3) 55 °C for 30 s (4) 72 °C for 30 s and final elongation at 72 °C for 5 min. Products were purified using a Zymoclean gel DNA recovery kit (Zymo Research, USA). Purified amplicons were barcoded using the NEBnext Multiplex oligos for Illumina indices. The indexed amplicon libraries were purified using the Agencourt® Ampure® XP bead protocol (Beckman Coulter, Indianapolis, Marion County, IN, USA). Library concentration was measured using Nebnext Library quant kit (New England Biolabs, Ipswich, MA, USA) and quality validated using Agilent 2100 Bioanalyser (Agilent Technologies, Santa Clara, CA, USA). The samples were pooled in equimolar concentrations and diluted to 4 nM based on library concentrations and calculated amplicon sizes. The library pool was sequenced on a MiSeqTM (Illumina, San Diego, CA, USA) using a MiSeqTM Reagent kit V3 600 cycles PE (Illumina, San Diego, CA, USA). The final pooled library was at 10 pM with 20% PhiX as control. 20 Mb of data of 2 × 300 bp long reads per sample were produced. The list of primers and sequences used for the detection of fungal species is provided in Table 1. 

Data analysis (characterisation) and identification of fungal species was performed using BLAST searches, GenBank, and Inqaba in-house developed (Figure 2) data analysis pipeline.

### 2.3. Soil Analysis

Soil samples were collected at the same study sites at Telperion Game Reserve as the leaves and caterpillars samples. Soil analysis was performed on 30 samples of *Burkea* soils representing the rhizosphere and another 30 samples of non-*Burkea* soils representing the non-rhizosphere soils. Soil samples (500 mg) were subjected to DNA extraction using a NucleoSpin Soil DNA kit (Mo Bio Laboratories, Carlsbad, CA, USA) according to the manufacturer’s instructions, and results were confirmed with agarose gel before sending for Polymerase Chain Reaction (amplification and cloning of DNA) and sequencing at Inqaba Biotechnology industry, Pretoria, South Africa as previously described [4].

## 3. Results

### 3.1. Higher Order Classification of the Microorganisms in the Caterpillars

The results of the study showed that the Fungal kingdom was most prevalent (96.78%), followed by an uninformative Kingdom (3.17%), which could not be classified and/or accurately identified under any kingdom. Bacteria and Plantae had the same percentages of 0.03 whilst Protozoa occupied the least percentages of 0.01, respectively, which yielded poor sequences; therefore, these were not accessioned or subjected to further analysis. 

#### 3.1.1. Family Classification of Ascomycota in the Caterpillars

*Pleurostomophora* dominated (60.08%), followed by *Trichocomaceae* (32.91%). The third and fourth families were uninformative and could not be assigned to any classification and were recorded at 6.08 and 0.45%, respectively.

#### 3.1.2. Species Classification

The species which took predominance was the fungi *Pleurostomophora richardsiae* (60%); the second dominant was *Aspergillus nomius* (32%) (Figure 3). 

Each fungal species detected and identified in *C. forda* was represented by a specific accession number and confirmed by NIH in the National Library Medicine at the National Centre for Biotechnology Information (Table 2).

An operational taxonomy unit (OTU) was done to indicate clustering and long reads to generate percentage identity of the species identified in *C*. *forda* caterpillars, produce more accurate and prediction fungal species as shown in Table 3.

The results of tree construction, replication and scale used in a phylogenetic tree indicating the probability and higher percentages of mean close relatedness of fungal species are Figure 4.

### 3.2. Classification of the Microorganisms in the Leaves

The results of the study showed that the Fungal kingdom was most prevalent (99.47%), followed by an uninformative Kingdom (0.46%), which could not be classified under any kingdom, and Protozoa had the lowest percentage of 0.7, which yielded low similarity; therefore, these were not accessioned or subjected to further analysis. 

#### 3.2.1. Phylum Classification

The leaves of *B*. *africana* were dominated by fungi, notably Ascomycota (94%), followed by an unknown phylum (5%), with other phyla, such as *Tracheophyta*, *Proteobacteria* and *Ciliophora,* at almost undetectable levels.

#### 3.2.2. Family Classification of the Ascomycota in the Leaves

*Pleurostomophora* (72%) was found to be the most prevalent family, followed by *Togniniaceae* (14%) and *Polyporaceae* (6.05%).

#### 3.2.3. Species Classification

The species which took predominance was the fungi *Pleurostomophora richardsiae* (72%); the second dominant was *Phaeoacremonium scolyti* (14%), as demonstrated in Figure 5.

Each fungal species detected and identified in the leaves of *B. africana* was represented by a specific accession number and confirmed by NIH in the National Library of Medicine of the National Centre for Biotechnology Information (see Table 4). 

An operational taxonomy unit (OTU) was done to indicate clustering and long reads to generate percentage identity of the species identified in *B*. *africana* leaves, produce more accurate and prediction fungal species as shown in Table 5.

The results of the tree construction, replication and scale used in a phylogenetic tree indicating the probability and higher percentages of mean close relatedness of fungal species are shown Figure 6 below.

## 4. Discussion

The two fungal species, *P. richardsiae* and *A. nomius* were identified with high prevalence from the *C. forda* caterpillars, which were collected from *B*. *africana* trees. Additionally, *P. richardsiae* was dominant in both the *C. forda* caterpillars (60%) and the leaves (72%) of *B*. *africana* trees, as shown in Figure 3 and Figure 5, respectively. Findings from a previous study reported that *Penicillium* sp. was the most prevalent fungal species in the *Burkea* soils, whereas it was absent in the non-*Burkea* soils, indicative of the important role of the fungal species in providing a supportive soil environment for the trees to survive [4]. The BLAST analysis could not identify all the Kingdom, Family, and fungal species accurately, resulting in uninformative classification and Protozoa detection. Protozoa were detected in the BLAST analysis, although the BLAST analysis was not using a prokaryote-specific database.

Carlucci et al. [27] showed that *Penicillium* sp. could be divided into two subgenera (*Penicillium* and *Aspergilloides*). *Penicillium* and *Aspergillus* are therefore regarded as sister genera due to sharing of a common ancestor and microbial divergence [28]. Similar findings were confirmed by Crous et al. [29]. The International Commission on Penicillium and Aspergillus (ICPA) met in Utrecht, the Netherlands, and discussed the implications of the single-name nomenclature on *Aspergillus* and *Penicillium* taxonomy [30]. The similarities have often been described between members of the two genera. Carlucci et al., Houbraken et al., Visagie et al. [27,31,32] reported that *Aspergillus paradoxus* produce conidial heads with a terminal vesicle reminiscent of *Aspergillus* yet belong to *Penicillium* subgenus *Penicillium.*

It was, however, expected to find *A*. *nomius* together with *Penicillium* in the soil, for them to act as growth-inoculant fungi. The collection of samples from the soil and caterpillars from different locations and in different years might therefore explain why different strains of the same fungus were collected, and therefore, it is put forward that *Aspergillus nomius* and *Penicillium* are related fungi with slight differences in genetic makeup as reported by Crous et al. [29]. The presence of *A. nomius* species detected in the caterpillars hosted by *B*. *africana* trees and *Penicillium in* the *Burkea*-soil (soil where *B*. *africana* trees grow), therefore, confirms the link between the caterpillars as a host of *Aspergillus/Penicillium* sp. fungus. It is therefore proposed that the caterpillar is the source of inoculation of *A. nomius/Penicillium* sp., which serves as a constant and continuous source of inoculum in the soil. [4]. Since the similarities in the two fungal species have been reported in previous research, the current results confirm that *Aspergillus* and *Penicillium* are sister genera/fungal species, as evidently shown in the phylogenetic tree constructed (see Figure 6).

This study reported the presence of various growth-promoting metabolites (GPM) in *Burkea*-soils due to the presence of *Aspergillus*/*Penicillium* sp., where the fungal composition was linked to the development of *B. africana* trees and is assumed to be responsible for creating a supportive environment for the natural establishment and survival of seedlings [4]. The abovementioned is based on the absence of a dominant fungal species never found in any of the non-*Burkea* soils, even though the soil collections were often performed a few meters apart. The presence of a dominant fungal species in all the *Burkea* soils is therefore important, as it necessitates a continuous inoculum of the soil, especially around the trees.

*Pleurostomophora richardsiae* is an emergent fungal pathogen that has been associated with esca and Petri disease in California [33] and caused vascular discolouration after field and glasshouse inoculations similar to that seen in Petri-diseased grapevines in South Africa [34]. It is a rare dematiaceous (dark-walled) fungus that was previously known as *Phialophora richardsiae* but has been recently renamed [35]. It was first isolated from a patient with a *phaeomycotic* cyst in 1968 [36] and is found in the soil, decaying wood and vegetation [37]. Levenstadt et al. [38] reported that *P. richardsiae* was dominant in the leaves of almond trees. It is also considered the most aggressive pathogen among several other fungi found in almond trees [38], and its aggressiveness may be related to the concentration level found in the leaves, which in turn causes severe mechanical damage during and after the caterpillars’ invasion. In the current study, *P. richardsiae* fungus was also found to be highly dominant in the caterpillars, as shown in Figure 3. 

The current study represents the first report of *P. richardsiae* to be the main fungal species in or on the leaves of *B. africana* trees. This is also the first study to report that *P. richardsiae* is also found to be prevalent in the *C. forda* caterpillars, which feed on the leaves of *B. africana* trees. The fungal species is, however, not a deadly pathogen to the tree, as there are no reports of the death of *B. africana* trees caused by an infestation by *C. forda* caterpillars, although they cause severe defoliation by feeding on the leaves.

It is therefore suggested that *P. richardsiae* lives inside or outside on the *B. africana* leaves, and it is proposed that it indirectly influences host location and oviposition behaviour of Castor semi-looper moths which lays eggs on the leaves/branches of *B*. *africana* trees, which later hatch into *C. forda* caterpillars. This was also reported by Olmo et al., Vannette et al., Ballhorn et al., Rasmussen et al. [39,40,41,42] who stated that fungi are known to be important mediators of plant-herbivore interactions. 

Furthermore, other studies conducted by Locke and Crawford, Fontaine et al. [43,44] suggested that *P. richardsiae* is involved in the release of plant volatiles. It is, therefore, also possible that *P*. *richardsiae* plays a major role in attracting Castor semi-looper moths by releasing plant volatiles as cues when searching for their host to lay the eggs on, as the start of a life cycle of the caterpillar recorded from November 2021–January 2022 as demonstrated in Figure 7. The results of the study suggest that *P. richardsiae* plays a mediating role in *B. africana*-moth/caterpillar interactions.

These caterpillars are collected fresh, killed, preserved by adding salt and dried, thereafter sold in the streets markets by street vendors, who are mostly women. The caterpillars are considered a delicacy, eaten as a side dish (after they are boiled and fried) with pap, normally known as vhuswa, which is a hard porridge made of ground maise. *Cirina forda* caterpillars are known to be a high source of protein [45].

The processing of the caterpillars has been shown to introduce significant changes in some of the nutrients. Decreases in the concentration of nutrients, such as sodium, potassium, iron, magnesium, zinc and copper, were found in processed caterpillars as compared to fresh caterpillars [7].

*Aspergillus nomius* is a ubiquitous group of filamentous fungi spanning over 200 million years of evolution [46,47,48,49,50,51]. *Aspergillus nomius* is an aflatoxin-producing member of Aspergillus section Flavi that shows a cosmopolitan distribution. It has been described so far as a human pathogen in a case of breakthrough pneumonia in a patient with acute myeloid leukaemia [52]. In parallel, *A*. *nomius* has also been isolated from single cases of keratitis after ocular injury and onychomycosis in otherwise healthy patients [53,54]. Among the over 185 aspergilli, there are several that have an impact on human health and society [53], including 20 human pathogens, as well as beneficial species used to produce foodstuffs and industrial enzymes [55,56,57].

Furthermore, *A. nomius* is exceptional among microorganisms in being both a primary and opportunistic pathogen, as well as a major allergen [58,59,60,61,62,63]. This is supported by the relationship between *C. forda* and *B. africana* trees, which has shown that the infestation intensity does not result in the death of *B. africana* trees, except for severe defoliation. *Aspergillus nomius* produce carcinogenic secondary metabolites known as aflatoxins [64,65,66,67,68] responsible for hepatotoxic and immunosuppressive properties in humans and other animals [68,69] which may render agricultural products unusable as feeds and can lead to significant economic loss [70]. Several human case of ocular infection by *A*. *nomius* also has been documented [71,72,73] and several aflatoxin outbreaks in humans, following consumption of contaminated grain, have been documented [74,75,76,77]. Its conidia production is prolific and so human respiratory tract exposure is almost constant [78,79]. Concurrently, *Aspergillus* in human CARD9 deficiency has been referred as a fungal agent that shows predilection for non-pulmonary sites with little impact on the lungs [52]. *Aspergillus nomius* has been reported from tree nuts [80,81,82,83,84] sugarcane [85,86,87,88] and on assortment of seeds and grain [89,90,91,92]. Originally, *A. nomius* was considered rare, however, numerous studies have indicated that *A. nomius* is widely distributed and might be of economic importance [93].

*Aspergillus nomius* is often associated with insects, such as alkali bees [89] and Formosan subterranean termites [78] and is frequently isolated from insects’ frass in silkworm-rearing houses in Eastern Asia, Japan, and Indonesia. [56,80,94,95,96] also reported that *A. nomius* is found in dead or diseased insects.

Crops infected by *A*. *nomius* are the main sources for establishing soil populations, especially when colonized plant material is deposited onto the soil [97]. It is suggested that dead caterpillar bodies which fails to pupate and are found scattered around *B*. *africana* trees as shown in Figure 8, could serve as soil inoculum of *Penicillium* sp. which was found to be highly dominant in the soil where *B*. *africana* grows successfully [4]. 

Figure 9 illustrate the plant-herbivore and fungal species interaction for the effective growth of *B*. *africana* trees.

In addition, the current findings further suggest that large amounts of frass/droppings which are excreted by the caterpillars after feeding on the leaves onto the soils surrounding *B*. *africana* trees which ultimately, decompose and later inoculate the soil, could likely be involved in enhancing the growth of *B. africana* seedlings, although further research is needed to confirm this. What could be seen as an attack through the infestation of *B. africana* trees by *C. forda* caterpillars, supposedly colonize the caterpillars with *A. nomius* and later fall to the ground, decay, and in the process becomes a primary inoculum in the soils where *B*. *africana* trees grows. In the absence of a continuous introduction of inoculum into the soil by *C. forda* caterpillars, the fungal species are probably not maintained in the soil and might explain why tree and seedling growth outside its natural environments have not been successful. The absence of *C*. *forda* caterpillars found to be highly prevalent with *A. Nomius* reported to be related to *Penicillium* sp. could mean different soil composition which will not be conducive and favourable for continuous growth of *B. africana,* thus ultimately causing a slow death of *B. africana* seedlings grown outside their natural environment as reported by Nemadodzi et al. [4]. The factors which contribute and influence the release of volatile compounds which serve as an attractant of Castor-semi looper moths to lay their eggs on *B. africana* trees, however, is still not known which calls for further research. 

Processing of the caterpillars before consumption forms part of indigenous knowledge, to probably remove most of the harmful contents from the caterpillars before consumed, although information on the removal of fungal species in processing has not been investigated yet. Preparation of the caterpillars before consumption, includes removal of the intestines, and the caterpillars are boiled, dried, and fried before eating. This might also explain that no adverse effects have been reported by consumers of these caterpillars, although *P. richardsiae* and *A. nomius* are present in these caterpillars that are consumed. This however warrant future research to determine the role of processing of the caterpillars in reducing or even eliminating the fungal species before consumption.

## 5. Conclusions

Growing *B. africana* trees outside their natural habitat have proven difficult, which is the main reason these trees are not found in nurseries and not commercialised although highly regarded as an ornamental tree. Based on the findings of the current study, it is suggested that two fungal species play an important and integral role in plant–herbivore interactions to ensure the survival of the tree in harsh and challenging environmental conditions. *Pleurostomophora richardsiae* which is present in the leaves and the intestines of the caterpillars, provides a link to the association of the caterpillars with *B. africana* trees. *A. nomius* (reported to be related to *Penicillium* found in *Burkea-soil)* found in the *C. forda* caterpillars, which invade *B. africana* trees is hypothesised to play a substantial role in the growth and establishment of *B. africana* trees by being the main, continuous, and primary soil inoculant through colonization of their dead bodies which ultimately plays a vital role of enhancing and influencing the growth of *B. africana* trees and seedlings. This further reveals the mutual relationship which exists between *C. forda* caterpillars and *B. africana* trees as a host and source of food with *C*. *forda* playing a role as primary soil inoculants. Future research should be conducted to confirm and identify the possibility of volatile organic compounds which are released from trees that serve as cues in attracting the Castor-semi looper moths. Both the fungal species *P. richardsiae* and *A. nomius* present in the caterpillars have been previously recorded as human pathogens. This might raise a concern regarding the consumers and future studies should demonstrate the effect of these fungi on the larval consumer population. The traditional preparation and processing methods might be removing most of the pathogens and lower the risk of pathogen intake, although this warrants further research.

## Figures and Tables

**Figure 1 microorganisms-11-01864-f001:**
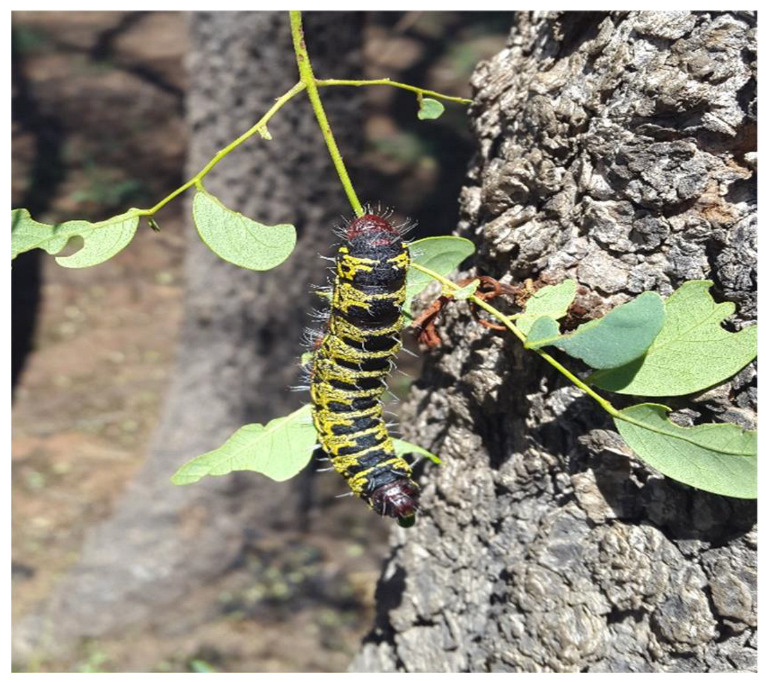
Caterpillar (*Cirina forda*) feeding on the leaves of *Burkea africana* (Photo taken by Nemadodzi L.E., 2016).

**Figure 2 microorganisms-11-01864-f002:**
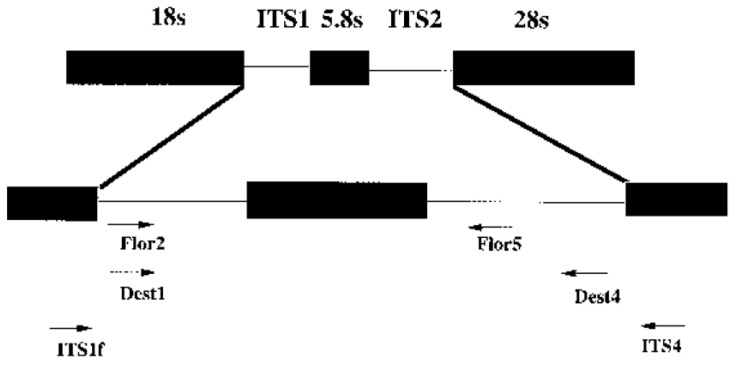
Illustration of universal primers used in fungal metagenomic analysis (Inqaba Biotechnology service provider, Pretoria, South Africa).

**Figure 3 microorganisms-11-01864-f003:**
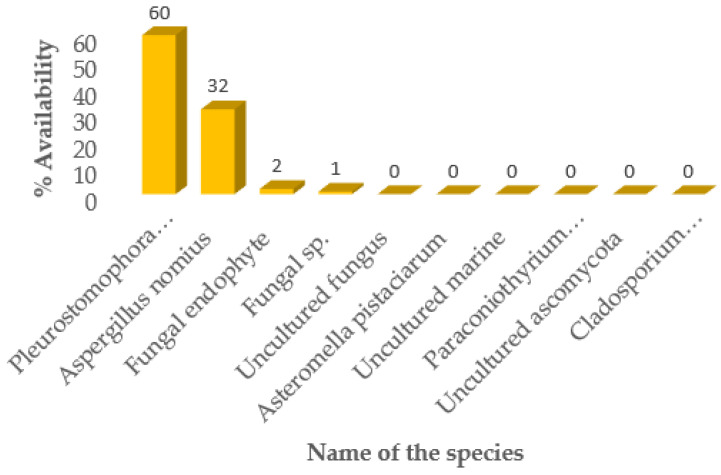
Fungal species identified in *C. forda* caterpillars indicated as % availability.

**Figure 4 microorganisms-11-01864-f004:**
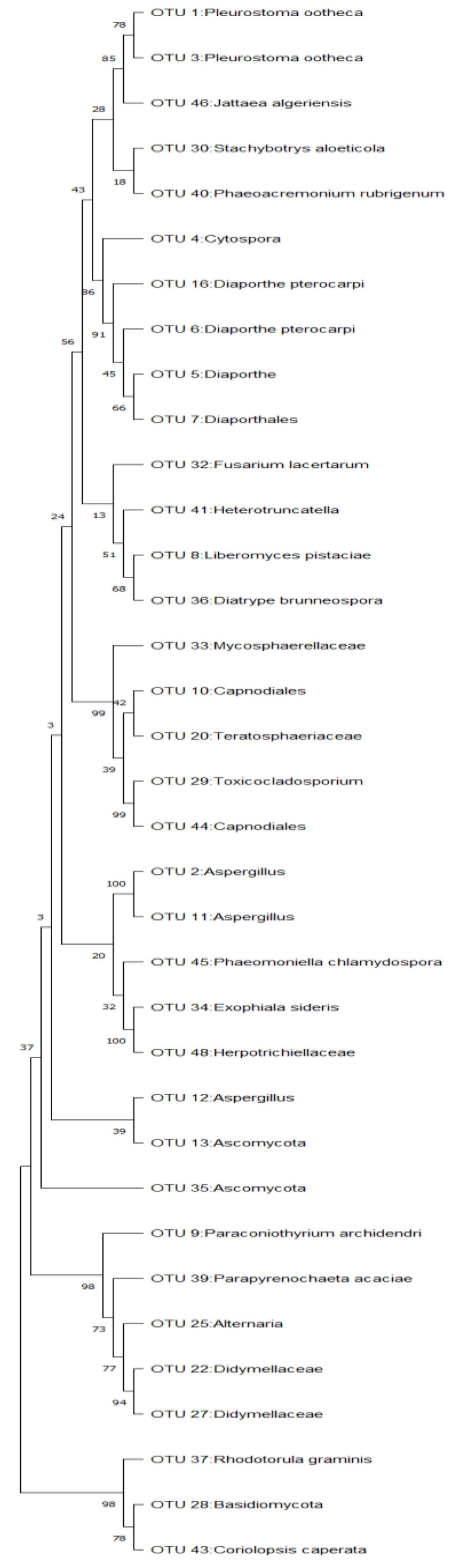
Phylogenetic tree constructed fungal species on *C*. *forda* caterpillars.

**Figure 5 microorganisms-11-01864-f005:**
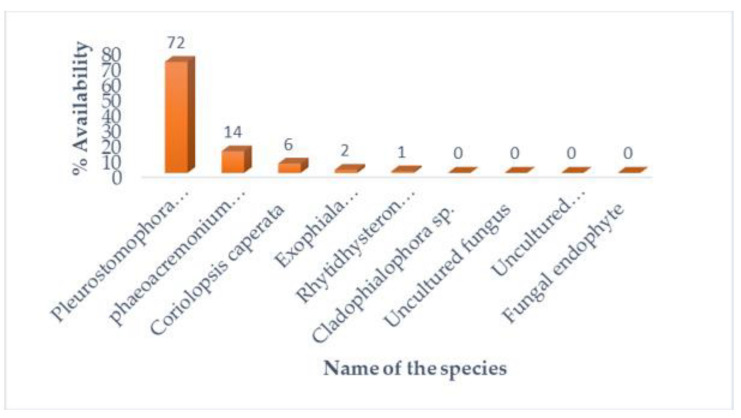
Fungal species identified in the leaves of *B*. *africana* trees indicated as % availability.

**Figure 6 microorganisms-11-01864-f006:**
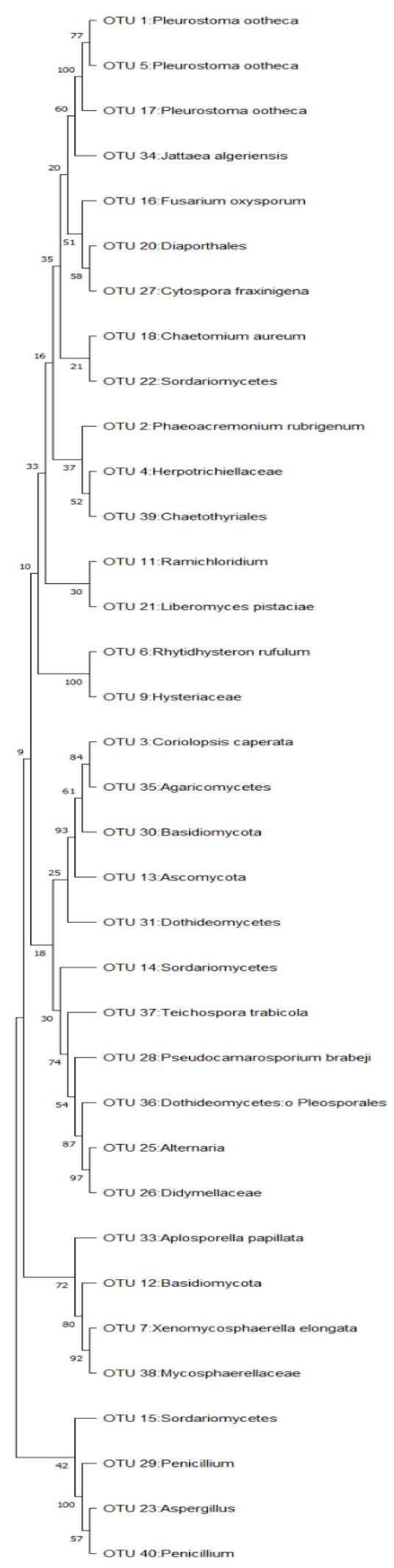
Phylogenetic tree constructed fungal species identified on *B*. *africana* leaves.

**Figure 7 microorganisms-11-01864-f007:**
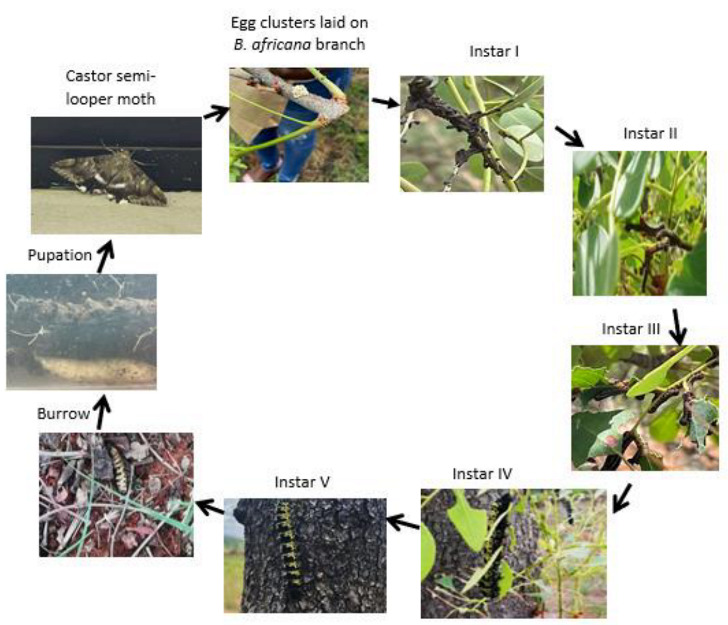
The life cycle of *Cirina forda* caterpillar (Nemadodzi L.E., November 2021–January 2022).

**Figure 8 microorganisms-11-01864-f008:**
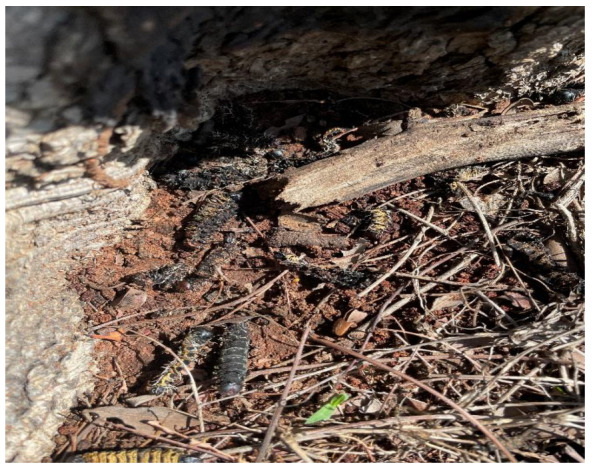
Dead *C*. *forda* scattered around *B*. *africana* tree (Nemadodzi L.E., December 2021).

**Figure 9 microorganisms-11-01864-f009:**
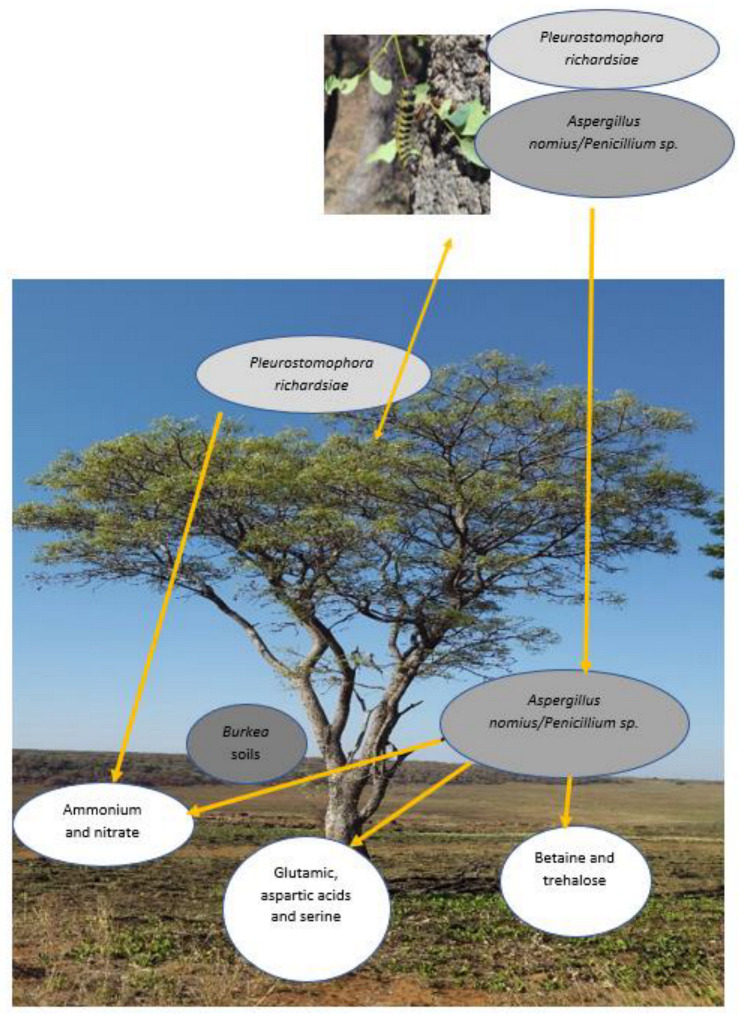
A schematic representation of the interactions between the fungi and the effects on the soil to ensure the survival of *B. africana* trees.

**Table 1 microorganisms-11-01864-t001:** List of primers used.

Primers	Sequences (3–5)
Truseq ITS 1F	TGACTGGAGTTCAGACGTGTGCTCTTCCGATCTCTTGGTCATTTAGAGGAAGTAA
Truseq ITS 4	ACACTCTTTCCCCACACGACGCTCTTCCGATCTTCCTCCGCTTATTGATATGC

**Table 2 microorganisms-11-01864-t002:** Hit per organism and their ITS accession number.

Organism/HIT	Cluster Size	Percentage	Genbank Accession #
*Pleurostomophora richardsiae*	16,413	60.07	KC341983.1
*Aspergillus nomius*	8993	32.91	KR905619.1
*Fungal endophyte*	771	2.82	HM537034.1
*Fungal* sp.	489	1.79	KC506340.1
*uncultured fungus*	252	0.92	KP167637.1
*Asteromella pistaciarum*	111	0.41	FR681903.1
*uncultured marine*	95	0.35	JX269272.1
*Paraconiothyrium hawaiiense*	75	0.27	KJ737370.1
*uncultured ascomycota*	48	0.18	KF060196.1
*Cladosporium cladosporioides*	34	0.12	KR012925.1
*Rhodosporidium babjevae*	8	0.03	KP732492.1
*Cytospora austromontana*	5	0.02	JN693510.1
*Coriolopsis caperata*	5	0.02	AB158316.1
*uncultured bacterium*	4	0.01	AB948531.1
*Pleurostoma ootheca*	3	0.01	AY725469.1
*Pestalotiopsis* sp.	3	0.01	KR012893.1
*Aspergillus oryzae*	2	0.01	KP794148.1
*Stachybotrys nephrospora*	2	0.01	AF081476.2
*Malassezia restricta*	1	0.00	JQ088233.1
*Chroococcidiopsis cubana*	1	0.00	HM630151.1
*uncultured eukaryote*	1	0.00	FJ176550.1
*Pestalotiopsis citrina*	1	0.00	KR065415.1
*uncultured bacteria*	1	0.00	HE611543.1
*Dothideomycetes* sp.	1	0.00	KM519276.1
*Aspergillus* sp.	1	0.00	KP686465.1
*uncultured gamma*	1	0.00	AY770726.1
*Chroococcidiopsis thermalis*	1	0.00	NR_102464.1
*No hits*	0	0.00	

**Table 3 microorganisms-11-01864-t003:** Taxonomy showing the assigned percentage identity of the evolutionary relationship of the fungi detected in the *C. forda*.

Feature ID	Taxonomy	Confidence
OTU_1	k__Fungi;p__Ascomycota;c__Sordariomycetes;o__Calosphaeriales;f__Pleurostomataceae;g__Pleurostoma;s__Pleurostoma_ootheca	0.9999509156065854
OTU_2	k__Fungi;p__Ascomycota;c__Eurotiomycetes;o__Eurotiales;f__Aspergillaceae;g__Aspergillus	0.9985353361019444
OTU_3	k__Fungi;p__Ascomycota;c__Sordariomycetes;o__Calosphaeriales;f__Pleurostomataceae;g__Pleurostoma;s__Pleurostoma_ootheca	0.9999969482027593
OTU_4	k__Fungi;p__Ascomycota;c__Sordariomycetes;o__Diaporthales;f__Valsaceae;g__Cytospora	0.854358736229371
OTU_5	k__Fungi;p__Ascomycota;c__Sordariomycetes;o__Diaporthales;f__Diaporthaceae;g__Diaporthe	0.8335552129523334
OTU_6	k__Fungi;p__Ascomycota;c__Sordariomycetes;o__Diaporthales;f__Diaporthaceae;g__Diaporthe;s__Diaporthe_pterocarpi	0.795273430121415
OTU_7	k__Fungi;p__Ascomycota;c__Sordariomycetes;o__Diaporthales	0.9999795871372799
OTU_8	k__Fungi;p__Ascomycota;c__Sordariomycetes;o__Xylariales;f__Xylariales_fam_Incertae_sedis;g__Liberomyces;s__Liberomyces_pistaciae	0.9182666027194447
OTU_9	k__Fungi;p__Ascomycota;c__Dothideomycetes;o__Pleosporales;f__Didymosphaeriaceae;g__Paraconiothyrium;s__Paraconiothyrium_archidendri	0.7832631665096351
OTU_10	k__Fungi;p__Ascomycota;c__Dothideomycetes;o__Capnodiales	0.9779387227439547
OTU_11	k__Fungi;p__Ascomycota;c__Eurotiomycetes;o__Eurotiales;f__Aspergillaceae;g__Aspergillus	0.9974969716266882
OTU_12	k__Fungi	0.999999999999996
OTU_13	k__Fungi;p__Ascomycota	0.7497696576994625
OTU_14	k__Fungi;p__Ascomycota;c__Dothideomycetes;o__Capnodiales;f__unidentified;g__unidentified;s__unidentified	0.8589742569616887
OTU_15	k__Fungi	1.0000000000000056
OTU_16	k__Fungi;p__Ascomycota;c__Sordariomycetes;o__Diaporthales;f__Diaporthaceae;g__Diaporthe;s__Diaporthe_pterocarpi	0.8165937605648326
OTU_17	k__Fungi	1.0000000000000115
OTU_18	k__Fungi	1.0000000000000049
OTU_19	k__Fungi	0.9999999999999927
OTU_20	k__Fungi;p__Ascomycota;c__Dothideomycetes;o__Capnodiales;f__Teratosphaeriaceae	0.8506001202469622
OTU_21	k__Fungi	0.9999999999999925
OTU_22	k__Fungi;p__Ascomycota;c__Dothideomycetes;o__Pleosporales;f__Didymellaceae	0.9192106392585596
OTU_23	k__Fungi	1.0000000000000016
OTU_24	k__Fungi	0.9999999999999964
OTU_25	k__Fungi;p__Ascomycota;c__Dothideomycetes;o__Pleosporales;f__Pleosporaceae;g__Alternaria	0.999990897152982
OTU_26	k__Fungi	0.9999999999999876
OTU_27	k__Fungi;p__Ascomycota;c__Dothideomycetes;o__Pleosporales;f__Didymellaceae	0.998632976176477
OTU_28	k__Fungi;p__Basidiomycota	0.7726061895145415
OTU_29	k__Fungi;p__Ascomycota;c__Dothideomycetes;o__Capnodiales;f__Cladosporiaceae;g__Toxicocladosporium	0.9999991617301468
OTU_30	k__Fungi;p__Ascomycota;c__Sordariomycetes;o__Hypocreales;f__Stachybotryaceae;g__Stachybotrys;s__Stachybotrys_aloeticola	0.9911876600078486
OTU_31	k__Fungi	0.9999999999999845
OTU_32	k__Fungi;p__Ascomycota;c__Sordariomycetes;o__Hypocreales;f__Nectriaceae;g__Fusarium;s__Fusarium_lacertarum	0.9385539330711409
OTU_33	k__Fungi;p__Ascomycota;c__Dothideomycetes;o__Capnodiales;f__Mycosphaerellaceae	0.9992143068624603
OTU_34	k__Fungi;p__Ascomycota;c__Eurotiomycetes;o__Chaetothyriales;f__Herpotrichiellaceae;g__Exophiala;s__Exophiala_sideris	0.9988522177150717
OTU_35	k__Fungi;p__Ascomycota	0.8032801838524619
OTU_36	k__Fungi;p__Ascomycota;c__Sordariomycetes;o__Xylariales;f__Diatrypaceae;g__Diatrype;s__Diatrype_brunneospora	0.9999959405293852
OTU_37	k__Fungi;p__Basidiomycota;c__Microbotryomycetes;o__Sporidiobolales;f__Sporidiobolaceae;g__Rhodotorula;s__Rhodotorula_graminis	0.9826119507998325
OTU_38	k__Fungi	0.9999999999999927
OTU_39	k__Fungi;p__Ascomycota;c__Dothideomycetes;o__Pleosporales;f__Pleosporales_fam_Incertae_sedis;g__Parapyrenochaeta;s__Parapyrenochaeta_acaciae	0.9998214149691789
OTU_40	k__Fungi;p__Ascomycota;c__Sordariomycetes;o__Togniniales;f__Togniniaceae;g__Phaeoacremonium;s__Phaeoacremonium_rubrigenum	0.8511770103314499
OTU_41	k__Fungi;p__Ascomycota;c__Sordariomycetes;o__Xylariales;f__Sporocadaceae;g__Heterotruncatella	0.8372277516833296
OTU_42	k__Fungi	0.999999999999982
OTU_43	k__Fungi;p__Basidiomycota;c__Agaricomycetes;o__Polyporales;f__Polyporaceae;g__Coriolopsis;s__Coriolopsis_caperata	0.7238326239885736
OTU_44	k__Fungi;p__Ascomycota;c__Dothideomycetes;o__Capnodiales	0.9999999997314887
OTU_45	k__Fungi;p__Ascomycota;c__Eurotiomycetes;o__Phaeomoniellales;f__Phaeomoniellaceae;g__Phaeomoniella;s__Phaeomoniella_chlamydospora	0.8528244306902385
OTU_46	k__Fungi;p__Ascomycota;c__Sordariomycetes;o__Calosphaeriales;f__Calosphaeriaceae;g__Jattaea;s__Jattaea_algeriensis	0.8490981193860933
OTU_47	k__Fungi	1.0000000000000087
OTU_48	k__Fungi;p__Ascomycota;c__Eurotiomycetes;o__Chaetothyriales;f__Herpotrichiellaceae	0.9057961832384308

**Table 4 microorganisms-11-01864-t004:** Hit per organism and their ITS accession number.

Organism/HIT	Cluster Size	Percentage	Accession #
*Pleurostomophora richardsiae*	55,201	72.73	KC341983.1
*Phaeoacremonium scolyti*	11,363	14.97	KC166687.1
*Coriolopsis caperata*	4594	6.05	AB158316.1
*Exophiala oligosperma*	2240	2.95	KT323978.1
*Rhytidhysteron rufulum*	1010	1.33	KJ787018.1
*Cladophialophora* sp.	511	0.67	AB986422.1
*uncultured fungus*	321	0.42	AB615469.1
*uncultured cryptodiscus*	225	0.30	KP323396.1
*Fungal endophyte*	126	0.17	KP335506.1
*Exophiala* sp.	96	0.13	HQ452316.1
*Pleurostoma ootheca*	76	0.10	AY725469.1
*Fusarium equiseti*	45	0.06	JN596252.1
*Dothideomycetes* sp.	40	0.05	AB986427.1
*Chaetomium aureum*	28	0.04	KC215131.1
*Pseudolachnella complanata*	7	0.01	AB934078.1
*Polyporales* sp.	6	0.01	JQ312175.1
*Coriolopsis* sp.	2	0.00	KJ612041.1
*Alternaria* sp.	1	0.00	KT186141.1
*Phaeothecoidea melaleuca*	1	0.00	HQ599594.1
*Aspergillus brasiliensis*	1	0.00	KM491891.1
*Predicted: mesocricetus*	1	0.00	XM_013111494.1
*Sporobolomyces griseoflavus*	1	0.00	AB038105.1
*Readeriella eucalypti*	1	0.00	GQ852781.1
No hits	o	0.00	None

**Table 5 microorganisms-11-01864-t005:** Taxonomy showing the assigned percentage identity of the evolutionary relationship of the fungi detected in the leaves of *B. africana*.

Feature ID	Taxonomy	Confidence
OTU_1	k__Fungi;p__Ascomycota;c__Sordariomycetes;o__Calosphaeriales;f__Pleurostomataceae;g__Pleurostoma;s__Pleurostoma_ootheca	0.9999969482027593
OTU_2	k__Fungi;p__Ascomycota;c__Sordariomycetes;o__Togniniales;f__Togniniaceae;g__Phaeoacremonium;s__Phaeoacremonium_rubrigenum	0.8511770103314499
OTU_3	k__Fungi;p__Basidiomycota;c__Agaricomycetes;o__Polyporales;f__Polyporaceae;g__Coriolopsis;s__Coriolopsis_caperata	0.7238326239885736
OTU_4	k__Fungi;p__Ascomycota;c__Eurotiomycetes;o__Chaetothyriales;f__Herpotrichiellaceae	0.9057961832384308
OTU_5	k__Fungi;p__Ascomycota;c__Sordariomycetes;o__Calosphaeriales;f__Pleurostomataceae;g__Pleurostoma;s__Pleurostoma_ootheca	0.9999509156065854
OTU_6	k__Fungi;p__Ascomycota;c__Dothideomycetes;o__Hysteriales;f__Hysteriaceae;g__Rhytidhysteron;s__Rhytidhysteron_rufulum	0.8900994594938357
OTU_7	k__Fungi;p__Ascomycota;c__Dothideomycetes;o__Capnodiales;f__Mycosphaerellaceae;g__Xenomycosphaerella;s__Xenomycosphaerella_elongata	0.9001828297722754
OTU_8	k__Fungi;p__Ascomycota;c__Lecanoromycetes;o__Ostropales;f__Stictidaceae;g__Cryptodiscus;s__unidentified	0.9998687556318772
OTU_9	k__Fungi;p__Ascomycota;c__Dothideomycetes;o__Hysteriales;f__Hysteriaceae	0.9996172473675551
OTU_10	k__Fungi;p__Ascomycota	0.8535115882460728
OTU_11	k__Fungi;p__Ascomycota;c__Sordariomycetes;o__Xylariales;f__Xylariaceae;g__Arthroxylaria;s__unidentified	0.7298461518721538
OTU_12	k__Fungi;p__Ascomycota;c__Dothideomycetes;o__Capnodiales;f__Dissoconiaceae;g__Ramichloridium	0.999999545156917
OTU_13	k__Fungi;p__Basidiomycota	0.8619031783651352
OTU_14	k__Fungi;p__Ascomycota	0.9143642096803432
OTU_15	k__Fungi;p__Ascomycota;c__Sordariomycetes	0.7838726000734728
OTU_16	k__Fungi;p__Ascomycota;c__Sordariomycetes;o__Hypocreales;f__Nectriaceae;g__Fusarium;s__Fusarium_oxysporum	0.98492272949497
OTU_17	k__Fungi;p__Ascomycota;c__Sordariomycetes;o__Calosphaeriales;f__Pleurostomataceae;g__Pleurostoma;s__Pleurostoma_ootheca	0.9999851012858587
OTU_18	k__Fungi;p__Ascomycota;c__Sordariomycetes;o__Sordariales;f__Chaetomiaceae;g__Chaetomium;s__Chaetomium_aureum	0.9574886918513182
OTU_19	k__Fungi;p__Ascomycota;c__Dothideomycetes;o__Pleosporales;f__Pleosporales_fam_Incertae_sedis;g__Parapyrenochaeta;s__unidentified	0.9991075911643039
OTU_20	k__Fungi;p__Ascomycota;c__Sordariomycetes;o__Diaporthales	0.9999795871372799
OTU_21	k__Fungi;p__Ascomycota;c__Sordariomycetes;o__Xylariales;f__Xylariales_fam_Incertae_sedis;g__Liberomyces;s__Liberomyces_pistaciae	0.9182666027194447
OTU_22	k__Fungi;p__Ascomycota;c__Sordariomycetes	0.7313138037550708
OTU_23	k__Fungi;p__Ascomycota;c__Eurotiomycetes;o__Eurotiales;f__Aspergillaceae;g__Aspergillus	0.999191894043742
OTU_24	k__Fungi;p__Ascomycota;c__Dothideomycetes;o__Pleosporales;f__Cucurbitariaceae;g__Curreya;s__unidentified	0.8751371838086929
OTU_25	k__Fungi;p__Ascomycota;c__Dothideomycetes;o__Pleosporales;f__Pleosporaceae;g__Alternaria	0.999990897152982
OTU_26	k__Fungi;p__Ascomycota;c__Dothideomycetes;o__Pleosporales;f__Didymellaceae	0.9192106392585596
OTU_27	k__Fungi;p__Ascomycota;c__Sordariomycetes;o__Diaporthales;f__Valsaceae;g__Cytospora;s__Cytospora_fraxinigena	0.934738163030393
OTU_28	k__Fungi;p__Ascomycota;c__Dothideomycetes;o__Pleosporales;f__Didymosphaeriaceae;g__Pseudocamarosporium;s__Pseudocamarosporium_brabeji	0.9445141771039463
OTU_29	k__Fungi;p__Ascomycota;c__Eurotiomycetes;o__Eurotiales;f__Aspergillaceae;g__Penicillium	0.998645369895211
OTU_30	k__Fungi;p__Basidiomycota	0.9219400198746588
OTU_31	k__Fungi;p__Ascomycota;c__Dothideomycetes	0.7100930622298849
OTU_32	k__Fungi;p__Ascomycota;c__Dothideomycetes;o__Pleosporales;f__Sporormiaceae;g__Sporormiella;s__unidentified	0.9652105615605318
OTU_33	k__Fungi;p__Ascomycota;c__Dothideomycetes;o__Botryosphaeriales;f__Aplosporellaceae;g__Aplosporella;s__Aplosporella_papillata	0.9079093067255156
OTU_34	k__Fungi;p__Ascomycota;c__Sordariomycetes;o__Calosphaeriales;f__Calosphaeriaceae;g__Jattaea;s__Jattaea_algeriensis	0.8490981193860933
OTU_35	k__Fungi;p__Basidiomycota;c__Agaricomycetes	0.9999999479732182
OTU_36	k__Fungi;p__Ascomycota;c__Dothideomycetes;o__Pleosporales	0.9907212343471081
OTU_37	k__Fungi;p__Ascomycota;c__Dothideomycetes;o__Pleosporales;f__Teichosporaceae;g__Teichospora;s__Teichospora_trabicola	0.9919360368035886
OTU_38	k__Fungi;p__Ascomycota;c__Dothideomycetes;o__Capnodiales;f__Mycosphaerellaceae	0.833939977809027
OTU_39	k__Fungi;p__Ascomycota;c__Eurotiomycetes;o__Chaetothyriales	0.9604725349305658
OTU_40	k__Fungi;p__Ascomycota;c__Eurotiomycetes;o__Eurotiales;f__Aspergillaceae;g__Penicillium	0.9941781575522258

## Data Availability

Data will be made available upon request.

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
