# Peer review of "A New Proposed Symbiotic Plant–Herbivore Relationship between Burkea africana Trees, Cirina forda Caterpillars and Their Associated Fungi Pleurostomophora richardsiae and Aspergillus nomius"

_microorganisms, 2023, doi:10.3390/microorganisms11071864_

Round 1

Reviewer 1 Report (Previous Reviewer 1)

Dear Authors

Thank you for answering the queries, but there are still some confusions in the text. Methods need further improvement to reach on a concrete conclusion. For example, protozoan sequences were reported in leaf samples which needs further clarification or resequencing. Moreover, the sample size is also small to conclude it. 

Thank you

Author Response

Dear Reviewer, 

The authors thank the reviewer for the valid concerns. 

 To be able to identify protozoa accurately, a novel rapid tool with the aid of a DNA barcoding marker will have to be developed as indicated in Line 199-204. In this study, a normal BLAST analysis which picked up protozoan sequences, although BLAST analysis was not using a prokaryote specific database. 

The reviewer is thanked for the concern. A metabolomics study was first conducted on the soil, and the metagenomics analysis was subsequently performed to understand the findings of the metabolomics analysis. In this study, 30 samples were used in the metabolomics study, which is in accordance with the study by Guo et al., 2012, who used 25 samples. The small sample size was therefore guided by the metabolomic clustering which showed the differences in soil around the trees (Burkea soils) in comparison other soils (non-Burkea soil) (see Line 75-79). In the metagenomics analysis, the number of the reads per sequence as explained in line 124 of the manuscript was important to obtain good data and to support metabolomics data.  

Reviewer 2 Report (Previous Reviewer 3)

In this manuscript, the author suggested the existence of a symbiotic relationship between the Burkea africana trees and Cirina forda caterpillars, where the caterpillars feeds on the leaves of the tree, when the caterpillars falls on the soil, it injects it with the A. nomius fungus, which indicates the existence of a symbiotic and / or synergistic relationship between the Burkea africana trees and Cirina forda caterpillars.

 Recognizing the validity of this hypothesis requires a lot of evidence that the current paper lacks.

The important point in this paper is that the local population feeds on the caterpillar that fed on the leaves of the tree. Thus, the population is exposed to the two fungi A. nomius and P. richardsiae, which are previously registered as human fungal pathogens, in addition to the fact that the A. nomius is a producer of carcinogenic aflatoxins.

From my point of view, I suggest that the authors amend the title to reflect the disclosure of the content of the tree and the caterpillars of the fungi. In view of the detection of two types of fungi previously recorded as human pathogens, future studies can be recommended to demonstrate the effect of these fungi on the larval consumer population.

In addition;

-       The phylogenetic analysis for the obtained results must be added

-       The author mentioned that the percentage of protozoa in the caterpillars was (0.01 %), while its percentage in leaves was (0.7 %).  It is known that, protozoa are commonly associated with animals. Do you have an explanation for that?

Typo errors

Line 62: correct” Pl. richardsiae” to “P. richardsiae

Line 64: delete the bracket “)”

Line 65, 66: correct “ Pl. richardsiae, Pl. repens, Pl. ootheca and Pl. ochracea” to be

                                      P. richardsiae, P. repens, P. ootheca and P. ochracea

Line 68: change the bracket format “{“ to “[“

Materials and Methods

Line 188, 239: correct” Pl. richardsiae” to “P. richardsiae

Line 210: write “penicillium” in italic

Add reference to section ; “2.2 Genomic DNA PCR and sequencing”

results

Fig. 5: correct “ Apsergillus” to “Aspergillus”

References

References NO. 70 , 90 are missed from text

minor editing of English language required

Author Response

Dear Reviewer, 

The reviewer is thanked for the positive critique, concerns and questions, below, is the point-by-point response to each. 

The title of the manuscript has been revised, corrected, changed and now reads ''A new proposed symbiotic plant-herbivores between Burkea africana trees, Cirina forda caterpillars and their associated fungi Pleurostomophora  richardsiae and Aspergillus nomius  as per the reviewer's advice.

Based on the nature and scope of the current study, only BLAST method was used in the identification of the species. However, the authors agree and appreciate the advice provided, follow-up and future studies will include a phylogenetic analysis. 

In this study a normal BLAST was performed which picked up protozoan sequences, although the BLAST analysis was not using a prokaryote specific database. To be able to identify protozoan pathogens accurately, a novel rapid tool with the aid of DNA barcoding marker will have to be developed.  

Typo errors

Line 62: corrected as advised and now reads P. richardsiae

Line 64: bracket) deleted. 

Line 65, 66: corrected and now reads P. richardsiae

Line 68: {changed to] as advised. 

Material and Methods

Line 188, 239: corrected and now reads P. richardsiae.

Line 210: Penicilium written in italic.

2.2 reference [26] added on genomic DNA PCR and sequencing on line 108 and added in the reference list as 26.

Results

Figure 5: spelling corrected and now reads aspergillus. 

References 

Reference 70 and 90 added intext as advised. 

Round 2

Reviewer 1 Report (Previous Reviewer 1)

Dear authors

although the manuscript has been improved, still there is much more to do. The phylogenetic analysis is essential to support the claims, however the methodology is superficial.

presence of protozoan sequences is still unclear which require additional experiments to confirm.

thank you

the English is ok, just need to check it again by a native speaker to improve long sentences.

Author Response

Dear Reviewer, 

The reviewer is thanked for the advice, although phylogenetic study is of importance, the authors believe that the revised manuscript has covered enough findings to be reported. Therefore, phylogenetic will be done as an add-on to the current findings and the findings reported thereafter on a separate manuscript. 

Protozoa occupied the least percentage of 0.01-0.07 which yielded poor sequences; therefore it was no accessioned or subjected to further analysis. 

We hope that you will find the response in order. 

Reviewer 2 Report (Previous Reviewer 3)

Dear authors with respect, this is not a good explanation, this reply makes the problem vaguer and delays the problem you raised in your results to the future works. We deal now about your current work in this manuscript, how you analyze your results and its interpretation.

And as far as I know, the Blast search is not a precise tool to identify a species. A Blast search and analysis, is mainly to detect the similarity in-between sequences, to identify a species, approach of ordinary and molecular tools should be applied.

And if you have already sequences, where the accession Nos?, and why you did not show the phylogenetic analysis of these sequences?!!

Author Response

Dear Reviewer, 

The reviewer is thanked for the concern, below is the response: 

The sequence together with the primers have been included as per your advice under 2.2, Table 1, line 135.

The accession numbers of fungal species detected on C. forda have been included (see Table2), line 188 as per your advice.

The accession numbers of the fungal species detected on the leaves of B. africana has been included in Table 3 in line 216 as per your advice. 

The reviewer is thanked for his advice, although the phylogenetic study is of importance, however, due to the limited scope of the current study, it was not done. The authors, believe that the findings reported on the revised manuscript should be reported. Phylogenetic study will therefore serve as an add-on to the current findings. 

We hope that the revised version will find favour in you. 

Round 3

Reviewer 1 Report (Previous Reviewer 1)

Dear Authors

Thank you for answering all the queries, i do not have any further question regarding the manuscript.

Thank you

Author Response

Dear Reviewer, 

The authors appreciate all the positive comments, advice, and questions raised in moulding the manuscript. 

Reviewer 2 Report (Previous Reviewer 3)

I thank the authors for their efforts in improving the manuscript, but

phylogenetic analysis are required, as I mentioned in my previous report.

 Minor editing of English language required

Author Response

Dear Reviewer, 

The author thanks the reviewer for the positive suggestion. The phylogenetic analysis which includes used, taxonomy (see Table 3 and Table 5), and phylogenetic trees (see Figure 4 and Figure 6) are included in the manuscript. 

The sequences used in the analysis are included under supplementary data. 

This manuscript is a resubmission of an earlier submission. The following is a list of the peer review reports and author responses from that submission.

Round 1

Reviewer 1 Report

Dear Authors

The present manuscript “A new proposed symbiotic plant-herbivore relationship between Burkea africana trees, Cirina forda caterpillars and their associated fungi” aimed to determine the possible symbiotic relationship between B. africana trees and the C. forda caterpillars and the mutual role played in ensuring survival of B. africana trees/seedlings in harsh natural conditions and low nutrients soils. Manuscript has interested findings for broad range of readers, although there are some small queries, please find them below.

Line 77- Please check the manuscript for language, for example “collecting of leaves” may be changed to collection of leaf samples.

Line 79- What does it mean the “newly flushed leaves”.

Line 92- “Both the leaves and caterpillars were sent to Inqaba Bio technical Industries, a commercial NGS service provider, for purifying and sequencing” Please explain this clearly.

Line 110- Please explain in detail the soil samples collection sites, weather conditions during sample collection and analytical methods. Overall the number of samples is less; I would suggest increasing the sample numbers to obtain more data which may facilitate for concrete conclusions.

The language need a revision by a native speaker or by an expert.

Author Response

Dear Reviewer,

The authors thank the reviewer for a positive critique. Below is point-by-point response to the questions, comments and concerns.

The manuscript has carefully been reviewed and the language improved as per track changes in the manuscript.

Line 77: Language revised and changed to collection of leaf and caterpillar (see line 87) 

Line 79: newly flushed leaves revised to newly developed leaves (see line 89). 

Line 92: The reviewer is thanked for the comment. The sentence was revised and rephased to provide clarity (Lines 102-130).

Line 110: the weather conditions are included now reads in 152-157. The analysis method was performed following the manufacture instruction as indicated in Lines 120-130. 

The study follows a previous study where a comprehensive metabolomic analysis were done on the soils. The metagenomics analysis was therefore additional to the findings that were already obtained and confirmed from other studies. The comment is, however, very valid as the current analysis shows that there is probably some variation in the fungal species found and a detailed phylogenetic study is envisaged where more samples will be collected and studied in detail. This has been included in the manuscript. 

Reviewer 2 Report

This work explores the mycobiome associated with C. forda caterpillars feeding on B. africana. The authors advance a fanciful hypothesis of a mutualistic relationship between these two organisms mediated by Aspergillus nomius / Penicillium sp. which proliferate in the insects and are delivered to the underlying soil through the caterpillar droppings and cadavers, providing soil with some quality essential for plant development. I find this hypothesis really uncircumstantial and incautious. Of course, it is possible that certain microbial associates of insects influence their development on selected host plants, which could be influenced in turn; however, such a conjecture must be based on solid evidences, which are completely lacking in the present case. In fact, besides taxonomic proximity, there is no relationship at all between A. nomius and Penicillium sp.; the genus Penicillium is ubiquituous, and many species are commonly represented in soil: how can authors consider a specific role without identifying a definite species? And what is the sense of the proposed transition A. nomius - Penicillium sp.? In my opinion the only interesting aspect which emerges in this study is the occurrence in the caterpillars of A. nomius and P. richardsiae, two species which are also known to be pathogenic to humans. This finding may deserve to be reported to the scientific community at least as a short note, considering that the caterpillars are exploited as a food source by the local populations.

The manuscript requires accurate revision for the English style and grammar.

Author Response

Dear Reviewer, 

The authors thank the reviewer for a detailed and positive reviews. The study reports ''a new proposed symbiotic plant-herbivore relationship between Burkea africana trees, Cirina forda caterpillars and the associated fungi".  Please take note that in this study, Penicillium was only found in the soil where B. africana trees grow (Burkea soils) versus the control, (soil where B. africana trees does not grow) herein referred to as non-Burkea soils.

It has been reported and confirmed that Aspergillus and Penicilliumare closely related by various studies at both morphological and in phylogenetic levels. After consultation with experts in the field on Aspergillus and Penicillium species, it was advised that the primers that were used, normally distinguished well between different species, but because of the similarities of these genera, that identification to species will need more in-depth analysis.  The authors did provide information in the current manuscript on the similarities of these genera which are clearly referenced.  The support for the proposed soil inoculation from the caterpillars is based on the comprehensive metabolomics study that was done on Burkea and non-Burkea soils. This study clearly indicated the dominance of fungal species in Burkea soils, not at all present in the non-Burkea soils, even though sometimes only a few meters apart. This has been shown for various soil samples collected from different sites, and the metabolomics analysis also supported this.  The presence of a highly dominant fungal species in both the caterpillars and the soil, and the reported similarity of both Aspergillus and Penicillium therefore provides the link between the proposed symbiotic relationship. The authors also acknowledge that the time difference of soil inoculation and variation in of fungal species that might be present in moths from different region might also affect the variation in fungal species. The   Authors, therefore, stats that the need for more in-depth phylogenetic studies to accurately determine the fungal species in the soil as well as in the caterpillars.  This has been included in the conclusion. 

It is true that A. nomius and P. richardsiae have been reported as human pathogens, and some species are also pathogenic in some plant species of wheat, maize, grapevine, oils seeds, nuts etc., which have been included in the current manuscript.  Why these fungal species are not pathogenic in these trees is not clear from this study yet and is being investigated in the current studies.  It is often reported that the processing of food makes them safe from the harmful compound and microorganisms that are present, which is very probable in this situation as well. The caterpillars are processed through various methods such as removal of intestines, boiling, frying and drying. Interestingly, the intestines contents were analysed for the fungal species which would not be present in the caterpillars used as food. Once again, this should be confirmed and warrant future studies on the safety and effect of processing on the presence of especially fungal species in the caterpillars. 

Reviewer 3 Report

Dear author

      Thank you for your work, it shows great potentiality into symbiotic relationships between Burkea africana trees, Cirina forda caterpillars and their associated fungi.

However, I have few comments/questions for you:

Abstract:

Lines 18-24; you reported that, the second most prominent fungal species in the caterpillars was Aspergillus nomius and it is proposed to inoculate the soil surrounding the trees with the fungus A. nomius. While in discussion (lines 157-159) you reported that, the two fungal species Pleurostomophora richardsiae and Aspergillus nomius were identified with high prevalence from the leaves of B. africana while P. richardsiae was dominant in the C. forda caterpillars. Explain it?

Introduction:

The introduction was devoid of any information about any of the fungi included in the article

Check it.

M&M:

a-     How did you make amplification (PCR conditions), please clarify this point in the manuscript.

b-    You run PCR using universal primers. Is it sufficient for identification and classification?

c-     As long as you don't use specific primers, at least you can do routinely cultural and morphological characteristics for identification

d-    Why you did not make phylogenetic analysis for the obtained results?

e-     Add reference for Figure 2

Results:

a-     Where is the accession numbers of the resulted sequences?

b-    Unknown kingdom?? How, explain

c-     Unknown family?? How, explain

d-    You decide the species level for some organisms, how you specify these organisms by your methods?

e-     Results obtained from leaves sequencing involving protozoa, how, explain?

f-     A symbiotic relationship has been proposed, the transmission of the fungus from the caterpillars to the soil, you used the same primers with the caterpillars and the soil: it was expected to discover Aspergillus in the soil as well “Regardless of the similarity between Aspergillus and Penicillium you discussed. Clarify?

g-    You did not get any results for bacteria, even though you used a universal primer for bacteria?

Discussion:

Revise well to be more clear and significant

References:

Reference No. 53 is missed from text

In reference list, names of many organisms need to be written in italic

Typo errors:

Correct figure numbers:

Line 130, 151 and 162 and 196

Line 96: correct “(ITS)1F and (ITS)4” to be “ITS1F and ITS4”

Figure 5, correct “Pleurostomorpha” to be “Pleurostomophora”

The attached file could be helpful

Author Response

Dear Reviewer,

The authors thank the reviewer for a detailed and positive review. Below are the responses to each question, comment and concerns.

Abstract: The reviewer is thanked for the comment. Line 205-207 has been revised, corrected and now reads as follows" the two fungal species Pleurostomophora richardsiae and Aspergillus nomius were identified with high prevalence from the Cirina forda caterpillars which had invaded B. africana trees''. Additionally, P. richardsiae was predominant in both the C. forda caterpillars (60%) and the leaves (72%) of B. africana trees as shown in Figure 3 and Figure 4 respectively. 

Introduction:

The reviewer is thanked for the comment. The missing information on fungal species have been added in lines 62-70. 

M&M:

a. The reviewer is thanked for the question. The amplification line was revised, rephrased and clarity provided in lines 102-130. 

b. The universal primer is normally sufficient as it is a well-established and accepted method. However, after further consultation, it was advised that due to the similarity between Aspergillus nomius and Penicillium genera, a more detailed phylogenetic study should be undertaken in future to accurately determine the specific fungal species.  The use of universal primers was sufficient at this stage to determine to determine fungal dominance of the specific fungal in the soil and caterpillars. 

c. The penicillium and Aspergillus are very similar as was also mentioned in the manuscript, which is remarkable at both morphological and genetic levels. The identification and characterisation of fungal species was done using BLAST searches (see lines 135-137) which is the most accurate method available at this stage of the study. 

d. A phylogenetic study was not in the scope of this study, but after consultation on the similarity of the two genera, it was advised that a phylogenetic study should be performed in the future. This has now been included in the conclusion of the manuscript.  Phylogenetic studies on the predominant fungal species are already in conception stage. 

e. The source has been added as per your suggestion. 

Results:

a. The reviewer is thanked for the question. The primers are included in the manuscripts on line 123-130 under Materials and methods.   

b. The reviewer is thanked for the comment. Unknown Kingdom now reads as ''uninformative kingdom'' (see line 169, line 174, line 187).

c. The sentences which initially had '' unknown family'' were revised, corrected and rephrased to read as ''uninformative family''.

d. The level of species organisms was assigned based on BLAST results. 

e. The reviewer is thanked for the question. The results showed high prevalence of 96.78% of Fungal kingdom, surprisingly, the least found was the Protozoa although in a very low of 0.01% percentage of 0.01%. Although the authors did not make further investigation on what could have led the presence in such a low availability, the authors opted to report the findings to create awareness to fellow researchers and help identify the gap for future studies.

f. The authors have provided clarification in lines 227-235. 

g. yes, there was no bacterial species found. 

The discussion was carefully reviewed, and improvements made as indicated in the track changes.

References: 

Ref No 53 intext has been added.

In reference list all the organisms have been revised and now in italics.

Typo error:

Line 130, 151, 162, 196 have been revised and the correct Figure added.

Line 109: (ITS)1F and (ITS)4 were revised and now reads ITS1F and ITS4 as per your suggestion now on lines 106-107.

Figure 5: Pleurostomorpha has been revised and now reads Pleurostomophora.

Round 2

Reviewer 1 Report

Dear Authors

The manuscript has been improved significantly and i do not have any further query.

Thank you

Reviewer 3 Report

There are still questions/comments that need clarification and amendment:
Results:
a-    Where is the accession numbers of the resulted sequences and phylogenetic analysis
b-    Unknown kingdom?? How, explain
c-    Unknown family?? How, explain
d-    You decide the species level for some organisms, how you specify these organisms by your methods?
e-    Results obtained from leaves sequencing involving protozoa, how, explain?
f-    A symbiotic relationship has been proposed, the transmission of the fungus from the caterpillars to the soil, you used the same primers with the caterpillars and the soil: it was expected to discover Aspergillus in the soil as well “Regardless of the similarity between Aspergillus and Penicillium you discussed. Clarify?
- I do not see any response to these remarks other than changing the word unknown with noninformative

 Minor editing of English language required